# Clinical Relevance and Advantages of Intradermal Test Results in 371 Patients with Allergic Rhinitis, Asthma and/or Otitis Media with Effusion

**DOI:** 10.3390/cells10113224

**Published:** 2021-11-18

**Authors:** David S. Hurst, Alan B. McDaniel

**Affiliations:** 1Department of Otolaryngology, Tufts University, Boston, MA 02111, USA; 2Department of Otolaryngology, University of Louisville, Louisville, KY 40202, USA; abmcdaniel0621@gmail.com

**Keywords:** allergic rhinitis, allergens, allergy immunotherapy, asthma, otitis media, sinusitis, skin prick test, allergy immunotherapy, intradermal allergy testing

## Abstract

Background: We evaluated the value of positive intradermal dilution testing (IDT) after negative skin prick tests (SPT) by retrospectively determining allergy immunotherapy (AIT) outcomes. Methods: This private practice, cohort study compared the relative value of SPT vs. IDT in 371 adults and children with suspected manifestations of allergy: chronic allergic rhinitis (AR), asthma and/or chronic otitis media with effusion (OME). The primary outcome measure was symptom resolution following immunotherapy, as determined by symptom severity questionnaires completed by patients before and after AIT. Results: Positive IDT identified 193 (52%) patients who would not otherwise have been diagnosed. IDT detected 3.7-fold more allergens per patient than SPT (8.56 vs. 2.3; *p* < 0.01). Patients positive only on IDT responded to AIT equally well as those identifiable by SPT, independent of allergen sensitivity (67% by SPT vs. 62% by IDT; *p* = 0.69, not significantly different). Conclusion: Intradermal titration can identify patients who will benefit from allergy immunotherapy more accurately than SPT. Outcomes analysis in 371 patients shows that IDT doubled their chance of successful treatment with no greater risk of therapeutic failure. Positive IDT, following negative SPT, is clinically relevant and offers superior sensitivity over SPT for detecting allergens clinically relevant to diagnosis of AIT-responsive atopic disease.

## 1. Introduction

The importance of IDT in the treatment of allergic disease: Many clinical studies over the past 75 years have shown that manifestations of allergy (rhinitis, chronic sinusitis (CS), asthma, eustachian tube dysfunction (ETD), and eczema) are associated with positive in vivo or in vitro tests for IgE-mediated hypersensitivity [1,2]. Allergy is common and it has an immense economic impact on school-age children and adults, their families and society [3]. Allergically mediated disorders can respond to immunotherapy, benefitting many patients by reducing their health care costs [4] and burden of illness, both present and future [5].

The decision to treat with allergy immunotherapy (AIT) is based on positive allergy tests. The success of AIT depends in large measure upon the ability of the chosen test method to correctly identify allergic individuals and their significant allergens [6]. Thus, the physician has incentive to employ the most reliable mode of testing.

Every allergist has tested clinically allergic patients whose skin tests are unaccountably negative. Furthermore, these “non-allergic” patients typically respond to treatment with antihistamines, decongestants, leukotriene inhibitors and/or steroids. This discrepancy implies that either the skin tests were falsely negative or the presence of localized, non-systemic allergic sensitization [7].

Both SPT and IDT were developed to improve upon the low specificity and poor reproducibility of scratch tests [6]. Out of concern that SPT may lack adequate sensitivity and understanding that IDT delivers larger doses of antigen, allergists can follow negative SPT with IDT when clinically indicated [8]. However, the benefit of performing IDT after negative SPT has long been debated [9].

It has proven difficult to demonstrate that adding IDT produces therapeutic gain. Any such benefit must be judged by patients’ superior response to AIT based on their particular test results [10,11,12]. Prior efforts to compare SPT and IDT have had insufficient populations to validly compare AIT outcomes [13,14].

This study of 371 patients was designed to test the hypothesis: the increased sensitivity of IDT compared to SPT is clinically relevant in patients with classic allergic diseases of AR/CS, asthma and/or ETD. The hypothesis will be validated if employing the information that IDT adds to SPT results is found to improve patients’ immunotherapy outcomes.

Objectives: This retrospective study was designed to assess the therapeutic value of diagnostic intradermal dilution testing (IDT) when patients’ skin prick tests (SPT) are negative.

## 2. Methods

The goal of this retrospective study in a solo, community-based practice is to compare AIT outcomes with the results of two methods of skin testing. Three hundred and seventy one allergic patients receiving immunotherapy were studied. Due to the presence of concurrent allergically mediated problems, 337 patients had allergic rhinitis (AR); 58 had asthma and 113 had chronic (≥3 months) obstructive eustachian tube dysfunction (ETD), as defined in a Clinical Consensus Statement [15].

The causal role of allergy in ETD and chronic otitis media (OME) was validated by an international panel of experts [16], who in their consensus report cited many of the first author’s publications. Most of the chronic ETD group has been previously reported, in which the role of allergy was confirmed [17,18].

In accordance with recent criteria [19], the diagnosis of AR was based on clinical symptoms and compatible physical findings, followed by positive skin tests [19]. Asthma was diagnosed by history, clinical examination and spirometry as indicated. All asthmatics also had either AR or ETD. The assessment of chronic ETD included symptoms of aural fullness, aural pressure and otalgia, often associated with hearing loss, or OME as documented by audiometry and tympanometry. Seventy-six of these 113 (69%) had previously been treated with a total of 182 tympanostomy tube placements (TTP) (87% of children, 66% of adults), a mean of 2.56 per patient.

All patients diagnosed with AR/CS, asthma and/or obstructive ETD who received AIT were included in this study. All groups were tested and treated identically as the groupings did not occur until this paper was written. Ethics approval was from the Franklin Memorial Hospital (Farmington, Maine) Committee on Human Experimentation. Informed consent was obtained from the patient or parents for both allergy testing and treatment.

### 2.1. Allergy Testing

All 371 patients were tested by the primary author using multi-dilution IDT, according to practice guidelines [20,21]. Included were a subgroup of 39 patients who had recently undergone SPT tests per board-certified allergists with few findings. Dissatisfied with their results, these patients sought a second opinion. Also included were a second subgroup of 68 with OME, as we previously reported [17].

All patients were tested by IDT for twelve locally appropriate antigens: *Dermatophagoides pteronyssinus*, *Dermatophagoides farinae*, cat, dog, American cockroach (*Periplaneta Americana*), grass (timothy or meadow fescue), tree (birch or oak), ragweed, goldenrod, lambs quarters, *Alternaria alternata* and *Cephalosporium acremonium* (Greer, Lenoir, NC, USA).

Children under age 10 were also tested for foods commonly found in their diet (dairy, egg, wheat, soy, chicken, potato, coconut, sugar, beef, corn and grapes) using molecular testing of RAST. The importance of food sensitivity in children was shown earlier: among 46 allergic OME patients, 45% were allergic only to inhalants while 55% reacted also to a food [22]. Testing by RAST and treatment of these food tests by diet elimination are not tabulated in our data.

The 2008 Updated Practice Parameters state that: “A suggested way of determining appropriate intracutaneous test concentrations is a serial end point titration regimen” [8] (a synonym for IDT). IDT is an intradermal bioassay that seeks both to identify allergens and to determine the lowest dose that triggers consistent hypersensitivity responses. Accordingly, IDT test antigens were serially diluted (5 fold, six times) from standardized extracts, usually 1:20 *w/v* (Greer, Lenoir, NC, USA). Our patients were tested with 4 mm intradermal wheals [23] and we used glycerin matched negative and positive (histamine) control tests [24]. Testing was begun at dilute antigen concentrations: dilution 5 (D5), 1:62,500 *w/v* or dilution 4 (D4), 1:12,500 *w/v* and progressed as indicated to stronger concentrations, with D2 (1:500 *w*/*v*) being the strongest dilution employed. Resulting wheal growth was measured 10 to 15 min after injection and the wheal size (in millimeters) was recorded [8,23].

The IDT end point is defined as the weakest dilution of allergen which produces a wheal that is 2 mm larger than the negative control wheal and very importantly, that is confirmed when the next stronger dilution results in a wheal that is at least another 2 mm larger [8,20,23]. Glycerin controls prevented misinterpretation of wheals that could lead to false-positive results (see discussion in [24]). Test results were used to inform each patient’s treatment options.

### 2.2. Relative Allergen Sensitivity

Patients were categorized according to the antigen concentration end points at which their skin responded, as well as by primary allergy diagnosis.

To learn more about the groups of SPT−/IDT+ and SPT+/IDT+ patients, test results following IDT were subdivided. As the low-sensitivity (SPT) cadre is definable only by IDT and because their response to AIT will test our hypothesis, we judged it important to carefully parse the testing data.

The low-sensitivity group was split into subsets A, B and C. In “A,” antigens reacted only at the strongest concentration, D2 (1:500 *w*/*v*). Group B patients had just one end point on D3 (1:2500 *w*/*v*) and all the other positives at D2. The “C” cadre had two or more confirmed reactions at D3 in addition to those at a D2 end point.

The high-sensitivity cadre was also divided into subgroups D and E. In D, patients had one antigen react on D4 (1:12,500 *w*/*v*), which is reliably positive by SPT. Group E had the most sensitive patients, who reacted to two or more antigens at D4 or weaker dilutions (Table 1).

### 2.3. SPT Status

The analysis of SPT status includes two data subsets. Thirty-nine patients were tested with SPT by one of 11 board-certified allergists within two years prior to enrollment. Records of these SPT procedures showed that testing conformed to American Academy of Allergy and Immunology guidelines [8] and the results were valid.

Other patients were tested only by IDT. Their SPT status was determined according to the following facts: the six-fold prick of the Multi-Test II (Lincoln Diagnostics, Decatur, IL) has a known sensitivity between IDT dilution 3 (D3, 1:2500 *w*/*v*) and dilution 4 (D4, 1:12,500) [6,25,26]. Additionally, the Updated Practice Parameters (SS 21) [8] state: “IDT is roughly equivalent to new skin prick tests only at dilutions ranging from D4 at 1:12,500 (*w*/*v*) to (D6) at 1:312,000 (*w*/*v*).” Therefore, a single-puncture SPT will reliably demonstrate a positive response to the same antigen dose that provokes a positive IDT end point on D4 (1:12,500 *w*/*v*) or weaker dilutions. Conversely, patients reacting only to greater doses of antigen (D3 or D2) are predictably SPT negative.

### 2.4. Exclusion Criteria

Patients who were completely negative for all IDT skin tests, including dilution D2, were not offered immunotherapy and were not entered into this study. Also excluded were those with craniofacial abnormalities, muscular dystrophy, history of previous cholesteatoma, or autoimmune disorder or those exclusively allergic to a food. All eligible patients were offered the same treatment options.

### 2.5. Immunotherapy

All patients who, following informed consent, received immunotherapy were enrolled into this study. Immunotherapy was indicated by the failure of avoidance and medications to adequately control allergy symptoms; the patient’s desire to reduce long-term use of medication; and positive IDT results. Study patients were all allowed to continue using antihistamines, inhaled steroids and antileukotrienes as needed.

Immunotherapy treatment extracts were formulated to all positive antigens to give the patient an initial dose of each positive antigen equivalent to 0.05mL of the “end point” concentration (defined above), as determined by his/her individual IDT results. After confirming the safety of the extract formula with a small dose (“vial test”), AIT was delivered by either subcutaneous injections (SCIT) once weekly or sublingual drops (SLIT) administered daily from individually prepared vials, made according to contemporary AAOA doctrine [23,27].

The dose of AIT was increased weekly, as tolerated. Either the maximum possible concentration of each allergen, or the largest tolerated dose was achieved within 4 months in all cases [28,29]. Patients were reevaluated every 5 to 6 months.

### 2.6. Outcome Measures

The primary outcome instrument was each patient’s self reported symptom score (parents graded their children’s results). A 10 point forced-choice Likert-type questionnaire was administered pre-treatment and after achieving AIT maintenance. Patients were asked: “One a scale of 1 (minimal) to 10 (terrible) how would you rate your current symptoms?” This question meets all criteria for designing Likert scales [30]. A Likert scale was chosen because of simplicity and ease of clinical use [31]. Responses were recorded in the clinical charts for later analysis. Observing the diminishing usage of medication and frequency of infections, particularly children’s OME, supported this analysis.

Of the 371 patients, 113 (30.5%) also provide the opportunity for objective outcomes analysis. This large minority had ETD with otologic consequences that were physically demonstrable on otoscopy, tympanometry and audiograms. Sixty-eight of these 113 patients with ear complaints had been reported previously [17]. This cadre is included here because, in addition to their Likert scores, the physical evidence they contributed attests to the benefits of AIT based on IDT (vs. SPT) and is relevant to the hypothesis.

### 2.7. Data Gathering and Analysis

Evaluation of IDT results in the 39 SPT-negative (SPT−) patients suggested segregating study patients into two groups, based on their test results. The 39 SPT+ and IDT-positive (IDT+) patients constituted the “low-sensitivity” group. After literature review and consultations, the senior author *also* added to this group patients tested by IDT who reacted only to antigen doses significantly greater than those prescribed by SPT and therefore are functionally SPT negative [8]. These dilutions include D3 (1:2500 *w*/*v.*) and D2 (1:500 *w*/*v*.).

The “high-sensitivity” group is composed of patients whose skin reacted to antigen doses that can be delivered by SPT, specifically D4 (1:12,500 *w*/*v*) and weaker [8]. Hence, they are termed SPT+/IDT+. The threshold at which the sensitivity of IDT differs from SPT is fundamental to testing our hypothesis.

Within these groups are five patient subsets, A through E, again divided for data analysis, according to the degree of skin reactivity on dilutional titration (Section 3.3, below). This subgrouping offers useful information but is not crucial to testing our hypothesis.

### 2.8. Statistics

We compared pre- and post-immunotherapy symptom improvement scores for patients, ranked by their degree of skin test sensitivity to allergens (Table 1) using the Chi Square test for differences. Statistical analyses were performed by a professional statistician using SAS 9.4 software (SAS^®^ Institute Inc., Cary, NC, USA).

## 3. Results

### 3.1. Demographics

Patients’ ages ranged from 6 to 73 years (227 females, average age 34.5 years; 144 males, average age 38.9 years). The subjects are demographically comparable and differ only in their degree of sensitivity on skin testing (Table 2).

There were 337 (91%) patients with allergic rhinitis, 109 (29%) with asthma and 113 (30%) with ETD or chronic otitis media with effusion (OME). There were equal numbers of SPT− (52%) and SPT+ (48%) patients (Figure 1). Forty three percent of patients had multiple allergies.

Percent of all patients with the same allergy grouped by relative allergen skin test sensitivity. Chi Square for different p values is shown. The improvement in the two groups, SPT−/IDT+ (Groups A, B, C) and SPT +/IDT+ (Groups D, E), is not statistically different—both for total patients and for any allergic disease. It was noted that there were relatively more OME patients in the lowest-sensitivity group, Group A (Figure 1), who only responded to testing at the most concentrated IDT strength of 1:500 *w*/*v*.

### 3.2. Direct Comparison of SPT and IDT Results

Of the 39 patients tested by SPT by their previous allergist, only eight (20.5%) had been offered AIT. They were retested by IDT within two years of their SPT. Thirty two presented with AR, 14 asthma, 14 chronic sinusitis and 7 had chronic ETD.

The results of testing the same twelve antigens by SPT and IDT among these 39 patients are graphically compared in Figure 2. Both IDT and SPT were positive in 19.6% and negative in 22% of tests. However, IDT found 3.7-fold more reactive allergens per patient than SPT (8.56 vs. 2.3) (Chi Square *p* < 0.01). SPT testing of these 39 patients for 12 antigens found 55 antigens positive and 413 negative.

SPT exhibited a sensitivity of only 16% of that of IDT. The validity of this assumption—and of our hypothesis—was soon confirmed, as IDT determined all 39 to be allergic and all received AIT.

### 3.3. IDT Results

On IDT, all 371 patients were displayed allergies to an average of 8.4 antigens. The number of allergens detected by IDT was significantly greater than that reported by SPT alone in both sensitivity groups. The increased number of positive antigens added by IDT is represented in Figure 3. Of the 3133 positive IDT antigen tests, only 12% (365) were detectable by SPT, i.e., being provoked by D4 (1:12,500 *w*/*v*) or weaker dilutions. Of the 88% (2764) of allergens detectable solely by IDT, 26% were positive only at the strongest dilution, D2 (1:500 *w*/*v*). Even among the 178 patients in groups D and E, IDT identified an additional 1235 allergens (6.9 per patient) (Figure 3).

Even in the high-sensitivity group (groups D and E), the majority of positive allergens were found with IDT, using larger doses of antigen than can be applied by SPT. Among these most sensitive patients, IDT identified an additional 6.9 allergens per patient (Figure 3). This “detection advantage” of IDT for the high-sensitivity (SPT+) group compared to the low-sensitivity (SPT−) group (Groups A, B, C) was significant (T test *p* = 0.004, 95% CI: (2.54, 0.506)). The clinical importance of this increased sensitivity is validated by our outcomes analysis.

### 3.4. Outcomes: Treatment with Allergy Immunotherapy

Allergy immunotherapy for four months reduced the average pre-treatment symptom scores of the 371 patients from 8.6 to 3.1, an overall improvement of 64% (Figure 4). We deem this a clinically meaningful improvement, since the World Allergy Organization (WHO) has stated that “the relative difference in the combined symptom score between active and placebo groups in AIT studies should be at least 20%” for significance, as based on quality of life (QOL) instruments [12]. The Likert scale scores and QOL instrument scores are similar in most cases, although QOL scores tend to have more statistical power [32,33,34]. 

The magnitude of improvement in the high- and low-sensitivity groups was compared. There is no significant difference between the 67% improvement among the most sensitive SPT+/IDT+ patients (groups D and E) and the observed 62% for the least sensitive SPT−/IDT+ patients (groups A, B and C) (Chi Sq. 0.03, *p* = 0.86; 95% Cl 0.048, 0.040) (Figure 4).

For each of the 5 subgroups A to E, with differing degrees of allergen sensitivity, improvements ranged from 57% to 71% (Table 1). We compared the outcomes of the two most different subgroups: A, who reacted only to the strongest antigen concentration and E, who reacted to the weakest doses of antigen. These “skin test response extremes” showed no significant difference in therapeutic response (Chi Sq. 3.73, *p* = 0.054; 95% CI: 0.004, 0.467).

### 3.5. Relative Allergen Sensitivity

Allergen sensitivity patterns of the 371 patients were similar among patients. Symptom improvement was also unrelated to allergy diagnosis. Table 4 exhibits the degree of symptom improvement amongst SPT−/IDT+ and SPT+/IDT+ patients for four combinations of clinical allergy diagnoses (AR, AR + Asthma, AR + OME, AR + Asthma + OME). These were not statistically different by Chi Square tests. Importantly, *in no allergy diagnostic category was the degree of skin sensitivity significantly related to outcome.*

### 3.6. Otic Symptoms Responded to AIT

Among the 371 patients, 113 (30.5%) had ETD with ear symptoms (Table 4, Figure 1), which accompanied their bouts of sinusitis or asthma. In this group, nearly half (53/113) were adults over age 20. This is consistent with the reported 15 to 48% prevalence of EDT among chronic rhinosinusitis patients [35].

Their problems were significant, not trivial: of these 113, 71 (65%) had from 1 to 10 sets of ventilation tubes, totaling 182 tubes or 2.6 per patient. In addition, 18 were post-adenotonsillectomy and 7 had previous adenoidectomy.

Sixty-eight of these 113 patients with ear complaints had been reported previously [17]. All 68 were displayed allergies to an average of 9 antigens (Table 5). This cadre is summarized here because, in addition to their Likert scores, they contribute physical evidence of the benefit conferred by AIT based upon IDT.

Of the 441 antigens detected among those OME patients who resolved on AIT 79% of positive antigens reacted only at D2 (1:500 *w*/*v*) while only 9% were detectable with D4 (1:12,500 *w*/*v*) or weaker dilutions.

The 60 patients who resolved their ear problems demonstrated both normalized or improved tympanograms and prolonged absence of recurrent effusions or otalgia following AIT treatment. Our conclusion that their improvement was a consequence of immunotherapy is based on the fact that the entire control group of 21 patients, of whom all had the same medical and surgical treatment except AIT, failed to resolve [17]. Following immunotherapy based on IDT, 60 patients (88%) had full resolution of their ear problems, including all 3 wet mastoids, all 14 patients with chronic middle ear effusions and all 19 with chronic type C tympanograms. They maintained resolution for four to seven years of follow up. The 8 who failed were all ≥33 years old (average age 55.7). Five of these failures were referred for possible eustachian tube dilation of which none were found to be candidates.

## 4. Discussion

### 4.1. Allergy Skin Tests–Sensitivity vs. Specificity

The results of this study support our hypothesis: the increased sensitivity of IDT compared to SPT is clinically relevant in patients with classic allergic diseases of chronic rhinitis, asthma and/or ETD.

The optimal therapy for an allergic condition is successful hyposensitization immunotherapy (AIT), which should render a patient symptom free and drug free. Jacobsen et al. clearly stated the value of AIT when they wrote: “Allergen specific immunotherapy is the only treatment that interferes with the basic pathophysiological mechanisms of the allergic disease and thereby carries the potential for changes in the long term prognosis of respiratory allergy. Subcutaneous Immunotherapy (SIT) should be recognized not only as first-line therapeutic treatment for allergic rhinoconjuctivitis but also as secondary preventive treatment for respiratory allergic diseases” and can prevent new onset asthma [36].

However, the success of AIT depends upon accurate allergy tests to identify clinically relevant allergens, which can then be treated. This requirement has proven to be a challenge, for the sensitivity and specificity of various testing modes are not sufficiently well defined to permit a unanimous choice.

Because of their higher diagnostic sensitivity for symptomatic allergy, skin tests are generally preferred over in vitro methods such as antigen specific IgE or total IgE [37], especially in food allergy testing [38,39]. Skin prick test results correlate well with symptoms and the specificity is satisfactory, although “patients with a low-sensitivity IDT often have the only positive results” [14].

Our data indicate that SPT does not provide sensitive results and it may miss many allergens. To improve upon the sensitivity of prick tests, many clinical allergists follow them with selected intradermal tests, as described in Bernstein; et al.’s 2008 Practice Parameters [8] (Ref [8] Supplement Pg. S6). This is performed at the risk of increasing false-positive results. Thus, the specificity, and therefore therapeutic value, of these additional tests is debated [9]. The results of this study support the utility of IDT when selected patients are SPT negative and validate the specificity of IDT at high antigen concentrations for “low-sensitivity” patients.

### 4.2. IDT Identifies Allergic Patients More Accurately than SPT

To compare the relative values of SPT results (low sensitivity, high specificity) with IDT results (high sensitivity, *uncertain* specificity), 371 patients were divided into two groups based on their SPT results. “High-sensitivity” patients (n = 178) had one or more allergens at sensitivity levels detected or detectable by SPT, while 193 “low-sensitivity” patients had none (Table 1 and Table 2).

Because all IDT-positive patients were offered AIT based on these results, the treatment outcomes between these “high-” (SPT+) and “low-sensitivity” (SPT−) groups shall define the value of the information added by IDT tests with stronger doses of antigen (Table 1 and Table 4).

There was no significant difference in the total percent improvement scores between the high- and low-sensitivity groups (Chi Sq. 0.032, *p* = 0.86%; 95% CI 0.048, 0.040) (Figure 4). Both groups responded well—and equally—to AIT. If the IDT results had been false positives, many patients would have been unnecessarily treated and thus, would have not improved.

If only some of the IDT results had been false positives (e.g., at D2, 1:500 *w*/*v*), the subset analysis of groups A, B and C may have revealed at which antigen concentration the tests became unreliable. This was not observed. Between the two extremes of skin sensitivity (Subgroups A vs. E, Table 1) there was no overall statistical difference in therapeutic response (Chi Sq. 3.73, *p* = 0.054; 95% CI: 0.004, 0.467). There was a greater difference in improvement as reported among the children (73% vs. 48%) that may have been a reflection of the initial symptoms as perceived among those children’s parents (Table 1). This may also have reflected the greater sensitivity to allergens of those in group E. We believe this result supports the legitimacy of IDT at 1:500 *w/v* (D2).

The success of AIT in this report is important. Jacobsen states: “Subcutaneous Immunotherapy (SIT) should be recognized not only as first line therapeutic treatment for allergic rhinoconjuctivitis but also as secondary preventive treatment for respiratory allergic diseases.” [36] He also showed, in his study with 7 years of follow up, that: “Patients who developed asthma among controls were 24/53 and in the SIT group 16/64. The odds ratio for no asthma was 4.6 95% CI (1.5 13.7) in favor of subcutaneous immunotherapy.” Instituting AIT not only treats allergic diseases of the unified airway, but can also actually prevent treated patients from developing asthma [40].

### 4.3. Relation of Allergy to OME

Some have questioned the causal linkage of atopy/allergic hypersensitivity and ETD/chronic otitis media with effusion. Although there are certainly many etiologies related to EDT” much evidence supports this association. Further discussion of relevant medical literature and our findings follows and will support this relationship.

Because of the wide distribution of immune effector cells, allergic reactions can affect any organ, including the eyes, nose and sinuses, middle ears and mastoids, pharynx and larynx, lower airway, skin and gut. Indeed, there are many similarities in the pathophysiology of allergic rhinitis, chronic sinusitis, asthma and ETD. It is established that all areas of the respiratory tract are capable of mounting an inflammatory response identical to each other. The concept of the “unified airway” closely links IgE-mediated hypersensitivity with allergic rhinitis/chronic sinusitis (AR/CS), asthma and otitis media with effusion (OME) [1,2,41,42,43].

Much evidence has been amassed validating a strong association of allergy with ETD [44,45,46,47,48], beyond the obvious embryological derivation of the mastoid sinus and its histologic mucosal homology with the other sinuses [16]. Studies have thus far established that: (1) based on objective allergy testing, the majority of OME patients are atopic [45]; (2) histology shows that all the mediators necessary for a Th2 allergic response are present in the middle ear [48,49,50,51]; (3) per the 2016 guidelines, the middle ear is part of the unified airway, and “like other parts of respiratory mucosa, the mucosa lining the middle ear cleft is capable of an allergic response” [16]; (4) patients’ chronic middle ear disease partially or completely resolves with AIT based on intradermal testing results [17]; (5) meta-analysis suggests a strong correlation between AR and OME among children [2], and that risk factors for chronic otitis media finds allergic rhinitis as a prevalent condition ranging from 24 to 89% [52], although, as we report, not all patients with ETD suffer from AR, and vice versa [44].

Because none of these conditions is exclusively caused by allergy, the importance of correctly identifying this underlying pathophysiology is crucial to instituting proper therapy.

Experimental provocation challenge exposure to allergen has consistently caused a dose dependent decrease in ET patency, regardless of whether a seasonal (ragweed) or perennial (dust mite) allergen was used [53]. Epidemiological evaluation of the records of 2.4 *Billion* pediatric visits found allergy to be associated with a 2 to 4.5 fold increased incidence of OME as compared to the non-allergic state [54]. As determined by allergy testing, the reported incidence of allergy being related to ETD and/or OME ranges from 15% to 93% in pediatrics and up to 35% among adults. This wide variation in incidence could be related to differences in testing methods [49] as demonstrated by our data.

The objective and controlled findings of improvement amongst our chronic ETD cadre further endorse the value of immunotherapy [17,22]. Of course conversely, their response to AIT also confirms the significant causal association of allergy to middle ear disease.

De Corso et al. provides an in depth explanation of physiopathology factors linking allergy to increased risk of middle ear inflammation [46]. His recent systematic review of 3010 papers found that “clinical evidence and analyses of biomarkers suggested that allergy may be linked to some phenotypes of otitis media and, in particular, to otitis media with effusion and acute re-exacerbations in children with middle ear effusion” [46]. Indicators of a Th2 -riven allergic response, mast cells [51] (Figure 5) with their mediator tryptase and degranulating eosinophils [49] are present in a majority of ears with chronic effusion as well as in the sinuses and lungs of allergic individuals.

## 5. Arguments for Adding IDT to SPT

The clinical indication for IDT in the face of a negative SPT is more closely examined here and further evidence is presented. In recent years, several papers have advocated that testing with SPT alone is completely adequate but our data oppose this.

### 5.1. Is SPT Sufficiently Accurate or May IDT Add Valid Information?

It has been argued that ID tests for allergens not detected by SPT offer no further clinically relevant information [55]. Proponents of SPT thereby select high specificity over high sensitivity. Calabria and Hagan have reported in detail the strengths and shortcomings of both SPT and ID tests [56]. He notes that SPT methods are not standardized and its results are poorly reproducible [8,26]. In addition, the few comparative studies using the less reactive pollens or molds have found SPT has low sensitivity [9,56].

As indicated by our “low-sensitivity” (SPT−/IDT+) group, SPT may incorrectly diagnose patients as non-allergic when in fact they who do respond to AIT. It has been suggested that patients with persuasive symptoms of AR but who are SPT negative should be diagnosed as having “non-allergic rhinitis with eosinophilia syndrome” (NARES), rather than low-sensitivity allergy [7]. Can this condition be simply an artifact of insensitive testing? Much evidence suggests this is the case:

The term NARES is applied to these SPT-negative patients [7] because their nasal cytology is consistent with local allergic inflammation. Patients diagnosed to have NARES due to a negative SPT nevertheless can react positively to antigens administered on nasal provocation challenges [57].

A large pediatric study included a group of “non-allergic rhinitis” patients—of whom 60% responded well to symptom based AIT for grass pollen or *Dermatophagoides pteronyssinus*, despite having been SPT negative [7]. This report supports observations that SPT has low sensitivity [9,56], which our study corroborated. However, the authors attributed the success of AIT not to low SPT sensitivity but to “Local (not systemic) Allergic Rhinitis” [7]. Our question is: Would adding IDT have provided useful information?

Practice guidelines [8,58,59] have recognized that IDT can be up to 1,000-fold more sensitive than SPT [60] because a significantly greater dose of antigen can be safely administered [28]. IDT is approximately equivalent to new skin prick tests only at dilutions ranging from 1:12,500 *w/v* (D4) to 1:312,000 *w/v* (D6) [8]. When it is important to detect an allergic etiology, a more sensitive test is preferred [59].

Calabria and Hagan stated: “For lower potency or non-standardized allergens, the ID skin test may identify a higher percentage of patients with lower levels of clinical sensitivity, and a positive test result may be more clinically relevant” [56]. Several studies in New York City have reported IDT results subsequent to negative SPT, finding among these “non-allergic” patients 20.8% [61] and 24% [62] additional positive antigens.

We agree with others that the ability of SPT to both diagnose allergy and detect the important allergens for effective AIT is limited [60]. We found that IDT alone, when used at 1:500 *w*/*v*, can physically introduce enough allergen to reliably detect most low-sensitivity allergies (Figure 2, Table 3). An earlier review by Gordon [6] also concluded that both single SPT and IDT (at dilutions from 1:12,500 *w/v* to 1:312,000 *w*/*v*) are equally able to detect high-sensitivity allergies.

### 5.2. Does IDT at Strong Dilutions Add Valid or False-Positive Results?

It is observed that increased sensitivity is associated with reduced specificity [58]. It is therefore argued that using *single*-dilution IDT may result in false-positive reactions. It is true that the more concentrated dilutions can produce false-positive tests by osmotic pressure and nonspecific irritation [8].

However, adequate precautions can be taken to minimize IDT false positives. Properly matched glycerin controls [24] are essential and easily employed. The method described above (Section 2.5) of assessing quantitative skin sensitivity with multiple dilutions per antigen to confirm the “end point” is highly reliable [23]. The chances of false-positive results are thus greatly reduced. These precautions are easily incorporated into routine practice.

### 5.3. The Outcome of AIT Validates the Accuracy of IDT

The outcome of immunotherapy should indicate the value of the test upon which the diagnosis and the formulation of extracts is based. The positive response of SPT-negative patients to AIT, whether adults, children or disease (Table 1, Table 2 and Table 4), is a convincing indication that their skin prick tests missed the diagnosis. However, comparing any two testing methods by assessing therapeutic outcomes has been difficult to accomplish. The few comparative studies using the less reactive pollens or molds have found SPT to be of low sensitivity [9,56], which our study corroborated (Figure 2, Table 3 and Table 5).

Prior attempts to measure AIT treatment results following IDT have been small studies with insufficient power [13,14]. Consequently, the value of adding IDT tests to SPT has not previously been convincingly demonstrated. We believe this study of 371 patients, which adds symptom relief scores to the previously reported objective improvement [17] provides more compelling evidence that IDT results are clinically valid and therapeutically beneficial.

### 5.4. IDT Offers Other Benefits over SPT

The office economy of SPT over IDT is often cited, but this is superficial, lacking a complete long-term analysis.

Kaffenberger et al. reviewed AIT results of patients tested by SPT, SPT and IDT or in vitro [14] and acknowledged that it took SPT-based AIT patients an average of 265 days longer to reach maintenance or 38 additional weekly visits (Ref. [14], Table 5). The advantages of SPT, including better patient comfort and economies of time, office expense and staff training [8,58], pale upon comparison with the cost of misdiagnosis, increased time to reach maintenance and consequential lack of effective treatment. More than half (193) of our 371 patients would have been denied a correct diagnosis and effective treatment without the addition of IDT Thus, the long-term cost difference favors IDT, which is superior to SPT for much the same reasons that CAT scans are superior to plain sinus films, despite being initially more costly and more time consuming.

## 6. Study Strengths

There are five major strengths to this study. The first is that this study was of a large group of 371 multi-symptomatic allergy patients who were assessed by comparing each patient’s own pre-treatment and post-maintenance therapy symptom scores (Table 1). The relevant measure by which allergy tests should be judged is the patient’s clinical response to AIT based on their test results [10,11,12]. To our knowledge, no large study has previously compared IDT and SPT methods using treatment outcomes based on the actual test results.

Pharmacotherapy for AR, when compared to placebo, has been reported to result in an average symptom improvement of 5% for leukotriene modifiers, 7% for antihistamines and 18% for intranasal corticosteroids [63], while median percentage changes from baseline total nasal symptom scores were 15% for placebo, 25% for antihistamines and 41% for intranasal steroids [64]. Analysis of three recent, large allergic rhinitis AIT trials based on SPT and total symptom scores, found that overall improvement ranged from only 10% to 29% [11]. In comparison, we report an even larger benefit from AIT averaging 64% symptom improvement based on IDT detection of allergens (Table 1 and Table 4; Figure 4).

The clinical benefit of AIT was also objectively demonstrated among our subset of patients in the OME group, 88% of whom experienced complete resolution of ETD symptoms [17].

Third, the possibility of false-positive results is one of the most frequently mentioned criticisms of IDT. A strength of this study is outcomes data convincingly addressing this concern. We compared patients at the extremes of sensitivity by examining the fraction of patients who failed to benefit from their immunotherapy in the SPT+ and SPT− groups. Both Groups A and E had large improvements (57% vs. 70%). These were not statistically different (Chi Sq. 3.73, *p* = 0.054; 95% CI: 0.004, 0.467) (Table 1). This was as expected, since the magnitude of the effect of AIT “is dependent on the severity of the allergic symptoms” [11] rather than on the degree of skin sensitivity.

Fourth, we had two sets of controls. (1) Among the ETD patients, 14 served as their own control. Although they had become free of effusion, otalgia or drainage during AIT, their symptoms or abnormal tympanogram recurred when they prematurely stopped their AIT. All again resolved after resuming AIT. (2) A second control group of 21 previously reported OME patients [17] who declined AIT also all failed to resolve their otalgia, conductive hearing loss, or OME.

There is no absolute way to evaluate allergy testing techniques other than by symptom resolution following allergy test-directed immunotherapy. Only 5.1% of SPT+/IDT+ patients reported treatment failure. There were 63 SPT-negative patients, all of whom, following IDT might be suspected of being IDT false positives. Their AIT failure rate of 5.7% was statistically identical to the 5.1% experienced by the 47 highly sensitive SPT+ patients (Chi Sq. 0.063, *p* = 0.80; 95% CI for difference: 0.040, 0.052). Given that the improvement rates were similar (Figure 4), the fact that the failure rates for both groups was also similar at 5% indicates that IDT results were *not* false positives.

Finally, a further study strength is the confirmation that AIT based on IDT improves compliance with immunotherapy. Kaffenberger et al. found that 78% of patients tested by IDT reached maintenance therapy, 30% more than those tested only by SPT (78.2% vs. 48.5%) [14]. This is compared to approximately 85% compliance in our study. It is possible that SPT misses key allergens, resulting in less effective immunotherapy and therefore patients do not perceive sufficient clinical benefit to continue investing their time and resources in continuing therapy.

## 7. Study Limitations

There were several study limitations. First, this study was neither randomized nor blinded, so there is the risk of bias, and the two control groups were self-selected.

Second, only 39 of the patients had been skin tested by both methods. The procedures of the practice studied do not include initial SPT; therefore, a direct comparison between SPT and IDT sensitivity was not available for the other 332 patients. Rather, we relied on extrapolating conventional accepted equivalency of IDT of <1:12,500 to the dilution of SPT [6,8,25,26]. This was supported by comparison of test results for those 332 to that of the 39 who did actually have SPT. The testing of those 39, although performed by AAAAI-certified physicians, may also have utilized antigens from a different company.

Third was the choice of a simple 10-point Likert scale to measure AIT outcomes. A Likert score questionnaire is a validated means of assessment. A Likert scale was chosen because of simplicity, validity, and especially ease of clinical use [30,31]. The absolute magnitude of improvement we observed using our symptom scale cannot be directly compared with literature values based on multi-question instruments. Although comparisons of Likert item scores and Quality of Life (QOL) instrument scores are similar in most cases, QOL scores tend to have more statistical power [32,33,34].

Simple data that are complete can be more useful than that which are complex but unfinished. Unlike our study, Kaffenberger measured AIT treatment outcomes following IDT using a multi-question QOL instrument, but he had problems with compliance. Only 14 (38%) of 37 patients who reached maintenance completed the full questionnaire [14], resulting in a small study with insufficient power for definitive conclusions.

Fourth, we lack a complete set of longitudinal data for all of the secondary outcomes. Pediatric patients have a natural history of resolution of EDT, but this study included 57 (50.4%) of patients with ETD aged > 15 years (Table 1, Figure 1), by which time most should have already resolved [17]. Furthermore, many of the EDT patients in this study had 3 or more sets of tympanostomy tubes prior to entry.

Fifth, one additional concern is that seasonal factors could have an impact on the outcomes depending on the group (SPT− versus SPT+). For instance, the test might be performed during an aeroallergen season and the outcomes of AIT could have been measured off-season. It is known that it is possible to test positive for trees or ragweed, or vice versa, even out of season; and conversely get relief when that season arrives. In fact, patients in both groups were tested throughout the year and some were treated over a period of several decades. That said, allergists are very concerned about testing someone who knows that they have trouble in a particular season such as late summer (ragweed) for that particular allergen during that season and the physician will postpone that test until out of season. The beauty of allergy immunotherapy is that seasonality is not a factor. 

Finally, it was difficult to distinguish differences between children and adults other than listing their testing and symptom responses in Table 1 and Table 2 and briefly noting some minor differences in Section 3.6, Section 4.3 and paragraph 4 of Section 7. Essentially, both adults and children demonstrated the advantages of IDT over SPT as both indicated a significant reduction in their symptom scores.

## 8. Conclusions

AIT depends upon accurate allergy tests to identify clinically relevant allergens that can then be treated. There is no better way to evaluate allergy testing techniques than by comparing symptom resolution with allergy test-directed therapy. This retrospective study evaluated symptom improvement among 371 AIT patients at maintenance-dose immunotherapy, compared with their own baseline symptoms recorded before starting therapy.

After six months of AIT, 94.3% of patients reported an average symptom improvement of 64%. IDT detected 3.7-fold more allergens per patient than SPT (8.56 vs. 2.3, *p* < 0.01). The implied false-positive skin test rate, suggested by treatment failure, was low, and statistically there was no difference by Chi Square test for SPT−/IDT+ failures (5.7%) as compared to SPT+/IDT+ failures (5.1%) (*p* > 0.05).

The data from these 371 patients with allergic diseases strongly support our hypothesis: positive IDT following negative SPT is clinically relevant and offers significant improvement in the ability to diagnosis AIT-responsive atopic diseases of allergic rhinitis/chronic sinusitis, asthma and/or ETD, especially in patients with low sensitivity (Groups A, B, C) whose reactivity to multiple allergens can only be detected with IDT.

IDT tests following negative SPT more than tripled the number of detected allergens, doubled the number of patients successfully treated with immunotherapy and increased the number of children diagnosed as being allergic by 58% (Table 1).

AIT produced nearly identical symptom improvement among low-sensitivity (SPT−/IDT+) (Groups A, B, C) patients who reported as much benefit from AIT as SPT+ patients (62% vs. 67%, *p* > 0.05, not different) (Table 1, Figure 3 and Figure 4).

Most of our patients were or would have been misdiagnosed by this insensitive method. They reported positive responses to immunotherapy, an opportunity the majority would have been denied if tested only by SPT. Importantly, patients found to be negative by SPT testing can never benefit from immunotherapy if they are incorrectly labeled as “non-allergic”. These data strongly support increased utilization of intradermal testing and invite additional clinical outcome studies.

## Figures and Tables

**Figure 1 cells-10-03224-f001:**
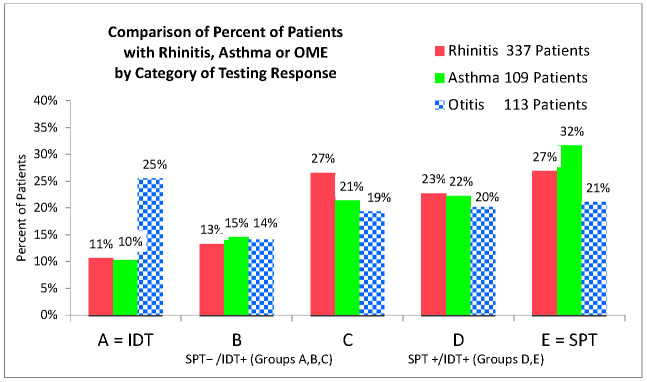
Comparison of Relative Frequency of Patients with Allergic Rhinitis, Asthma or OME by IDT Sensitivity.

**Figure 2 cells-10-03224-f002:**
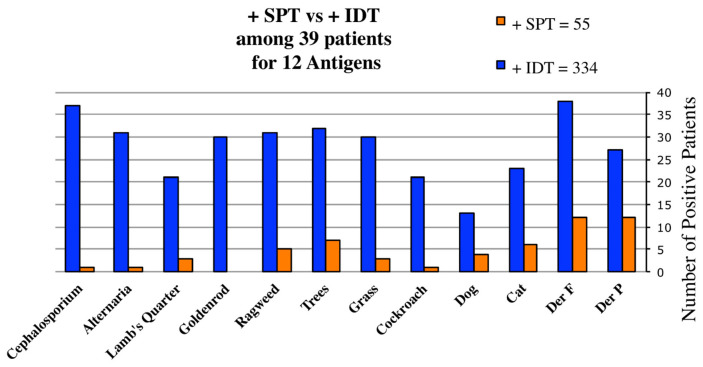
Comparative skin test results in 39 patients who initially had SPT and subsequently had IDT. The number of positive skin test reactions to each of 12 allergens following testing by both SPT and IDT. In comparison, their IDT results showed 334 positive and 134 negative. Individual antigen test comparisons for these 39 are shown in Table 3.

**Figure 3 cells-10-03224-f003:**
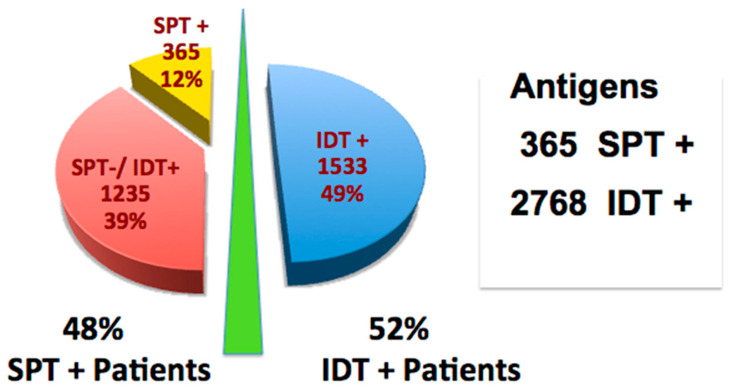
Benefit of adding IDT to SPT. Total of 3133 allergens discovered among 371 patients. IDT identified 1235 more treatable allergens among SPT+ patients (groups D and E) than SPT alone (365) (Chi Square *p* < 0.001, 95% CI: 0.47, 0.36; odds ratio 0.09; 95%CL: 0.05 to 0.15).

**Figure 4 cells-10-03224-f004:**
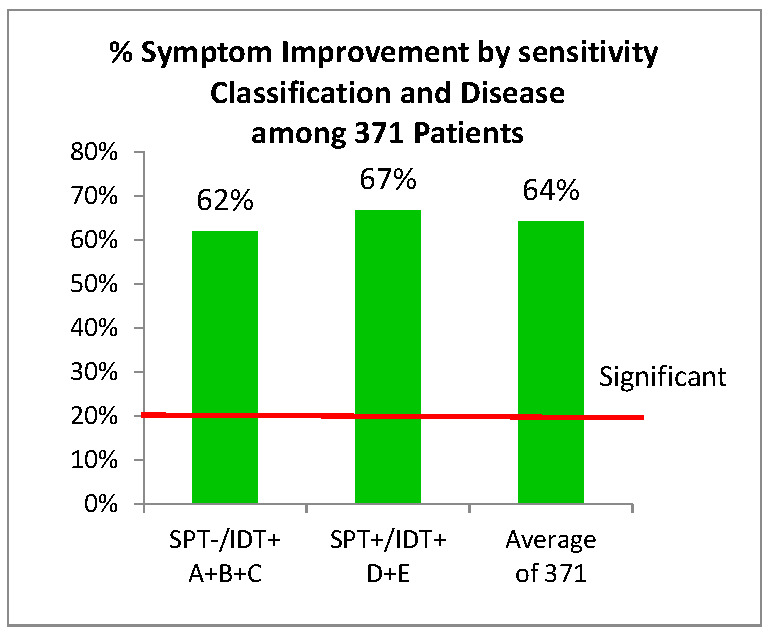
Symptom improvement by allergen sensitivity in 371 patients. A significant improvement is >20% [12]. The two groups are not statistically different.

**Figure 5 cells-10-03224-f005:**
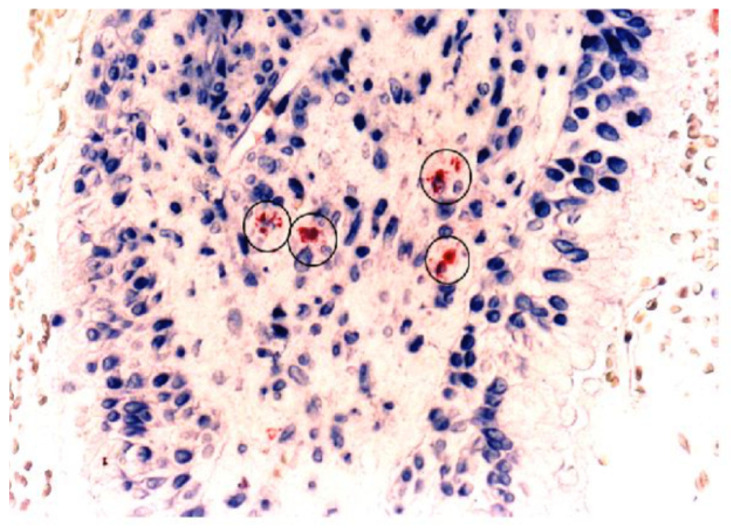
Mast Cells in Middle Ear Mucosa Biopsy [51] Antitryptase antibody (AA1) staining of mast cells (circled) (Adopted from: Hurst DS, Amin K, Sevéus L, Venge P. Evidence of mast cell activity in the middle ear of children with otitis media with effusion. *Laryngoscope*
**1999**, *109*, 471–477; with permission From Lippincott Williams & Wilkins, Inc.).

**Table 1 cells-10-03224-t001:** Immunotherapy Improvement: symptom scores by degree of allergen reactivity end points, before treatment and after reaching AIT maintenance among 371 patients.

		SPT−/IDT+				SPT+/IDT+	
Subgroup	A	B	C	Total	D	E	Total
Strongest +	Only D2	D2 + 1 D3	D2, D3	SPT−	D2, D3, +1 D4	D4	SPT+
Dilution(s)	1:500 *w*/*v*		1:2500 *w*/*v*		+1 D4	1:12,500 *w*/*v*	
**CHILDREN**							
Age 3–15	19	8	22	49	12	19	31
SYMPTOM SCORE							
Before AIT	7.3	8.7	8.5	8.3	8.5	9.2	8.97
After AIT	4.0	2.4	2.8	2.8	3.2	2.5	2.7
**% Improvement**	48%	72%	66%	68%	42%	73%	59%
Age 16–70	30	41	73	144	68	79	147
SYMPTOM SCORE							
Before AIT	8.6	8.5	8.8	8.4	8.6	8.5	8.6
After AIT	3.5	3.0	3.9	3.6	3.0	2.5	2.8
**% Improvement**	59%	65%	52%	61%	64%	71%	68%
**Total PATIENTS**	49	49	95	193	80	98	178
SYMPTOM SCORE							
Before AIT	8.34	8.47	8.75	8.58	8.62	8.58	8.61
After AIT	3.59	3.03	3.74	3.5	3.04	2.54	2.75
**% Improvement**	57%	64%	57%	60%	65%	70%	67%

Significant improvement is >20% [11]. Total % improvement SPT+ 67% vs. SPT −60% (Chi Sq. 0.032, *p* = 0.86; 95% Cl −0.048, 0.040) was not significantly different. Skin test responses extremes: improvement in Groups A vs. E (Chi Sq. 3.73, *p* = 0.054; 95% CI: −0.004, 0.467). Since *p* is >0.05, groups are not statistically different. The background colors are kept throughout the document to separate the two groups.

**Table 2 cells-10-03224-t002:** Demographics of patients with positive IDT and either negative or positive SPT.

	Low-Sensitivity Subgroups A, B, C	High-Sensitivity Subgroups D and E
Patients (%)	193 (52)	178 (48)
Total # Antigens * IDT+ SPT+	1490 0	1202 302
Avg. IDT+ Antigens *	7.9	8.9
Male # (%)	76 (53)	68 (47)
Female # (%)	117 (52)	110 (48)
Avg. Age in Years	39.6	40.9
Patients 3–15 Years Old	43 (22.2)	31 (17.4)
Patients 16–50 Years Old	75 (38.9)	79 (44.4)
Patients 51–75 Years Old	75 (38.9)	68 (38.2)

Note: The percentage of patients by age is % of total # patients. * The average number of antigens detected by IDT was significantly greater than that detected by SPT alone in both sensitivity groups. This detection advantage of IDT was even greater for the high-sensitivity (SPT+) group vs. the low-sensitivity (SPT−) group (*t* test *p* = 0.004, 95% CI: (−2.54, −0.506)).

**Table 3 cells-10-03224-t003:** Comparison of number of positive antigen test results from 39 patients tested by both SPT and IDT.

Antigen		IDT	D6	D5	D4	D3	D2	Total
Category	SPT	1:1000	1:312K	1:62K	1:12,500	1:2500	1:500	IDT
Der P	12	4	1		4	9	13	27
Der F	11	2	1	1	8	11	17	38
Cat	6	5			1	7	16	24
Dog	4	5					13	13
Cockroach	1				3	4	14	21
Grass	3	1			4	4	22	30
Trees	7	4				14	18	32
Ragweed	5	3		1	1	7	22	31
Goldenrod		1				4	24	28
Lamb’s Quarter	3	3				2	19	21
Alternaria	2	3			2	3	26	31
Cephalosporium	1	4			4	12	22	38
TOTAL	55	35	2	2	27	77	226	334
	SPT	Tests			IDT	Tests		

K = 1000. Der P = Dermatophagoides pteronyssinus; Der F = Dermatophagoides farinae.

**Table 4 cells-10-03224-t004:** Comparison of symptom improvement by allergen sensitivity and by type of allergic disease.

	Low Sensitivity		High Sensitivity			
	(Groups A+B+C)		(Groups D+E)			
	SPT−/IDT+ No Patients	% Improvement	SPT+/IDT+ No Patients	% Improvement	Total	Chi Square *p*-value
Allergic Rhinitis	94	58%	92	69%	186	0.57 = NS
Asthma	0		1		1	DI
OME	18		5		23	DI
AR + Asthma	33	63%	38	66%	71	0.32 = NS
AR+ OME	29	57%	24	66%	53	0.67 = NS
Asthma + OME	5		5		10	DI
AR + Asthma + OME	13	64%	14	65%	27	0.98 = NS
Total	193	60%	178	66.7%	371	0.86 = NS

Percent improvement from AIT of patients with the same allergy, grouped by relative allergen skin test sensitivity. SPT−/IDT+ (Groups A + B + C) vs. SPT+/IDT+ (Groups D+E). Groups A + B + C are least sensitive; Groups D + E are most sensitive. Chi Square for different *p* values is shown. The improvement in the two groups, (SPT−/IDT+) and (SPT+/IDT+), is not statistically different—both for total patients and for any allergic disease. DI = data insufficient. NS = not significantly different.

**Table 5 cells-10-03224-t005:** IDT end points for 12 allergens in 60 OME patients whose otologic symptoms resolved with immunotherapy [17].

		SPT+		IDT+			
CLASS + IDT	D6	D5	D 4	D3	D2	TOTAL	%
DER P		2	6		32	40	67
DER F	2	1	9		47	59	98
CAT				3	26	29	48
DOG				2	15	17	28
COCKROACH	1	1	1	7	29	39	65
GRASS		2	3	3	30	38	63
TREES			2	6	25	33	55
RAGWEED	2	1	4	3	31	41	68
GOLDENROD		1		7	28	36	60
LAMB’S QUARTER			1	3	17	21	35
ALTERNARIA				3	31	34	57
CEPHALOSPORIUM			1	15	38	54	90
TOTALS RESOLVED	5	8	27	52	349	441	
% OF + IDT	1.1%	1.8%	6.1%	11.8%	79.1%	100.0%	

SPT (yellow) reliably detects dilutions #6 through #4 [8]. SPT does not reliably detect dilutions #3 and #2 (green), although they are the majority of patients.

## Data Availability

The data presented in this study are available on request from the corresponding author, David Hurst. The data are not publicly available due to privacy of patient identity.

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
