# Peer review of "Clinical Relevance and Advantages of Intradermal Test Results in 371 Patients with Allergic Rhinitis, Asthma and/or Otitis Media with Effusion"

_cells, 2021, doi:10.3390/cells10113224_

Round 1

Reviewer 1 Report

line 32: rhinitis, sinusitis...are not allergy symptoms but allergy manifestations. 

Please discuss about allergy molecular tests 

Please discuss about NARES and LAR in clinical manifestations and in diagnostic process

line 66 ETD may have causes different from allergy

Please separate data of children from that of adults in the results and discuss the differences 

Author Response

Thank you for your observations. Hopefully, following your concerns we have improved the methods and results sections to your satisfaction

Reviewer 2 Report

In this study, the authors demonstrate the importance of using IDT versus SPT to diagnose allergy and identify the allergens to conduct immunotherapies to ultimately ameliorate the allergic symptoms or potentially cure allergy in patients.

My main concern is the lack of details regarding the types of AIT used in these 371 patients. Are there differences in AIT between the different groups? Some patients are allergic to 8 allergens. What allergen(s) were used in the AIT? Without these details, the description of the study is not complete.

It would be important as well that the authors comment on whether seasonal factors could have an impact on the outcomes depending on the group (SPT- versus SPT+). Have the groups been tested and treated at similar periods of the year. For instance, the test could be performed during an aeroallergen season and the outcomes of AIT could have been measured off-season. I realize that it must be difficult to include this information in the manuscript, but it could be mentioned as a limitation.

Note that some more complete information on the subjects, testing and treatments could be included in supplemental documents and spreadsheets.

Minor concerns:

  • Affiliations are not complete. There are no addresses. Who is the corresponding author?
  • Line 58, spell out “CS”.
  • When citing ref 17, please indicate “as we previously reported” instead of “reported elsewhere”. It is important to know whether previous work is from the same group or from another group.
  • In Table 1, A+B+C = 49, which is different from 43 in the “Total” column. Please check.
  • Line 171, “…6 months or sooner”. Anything more precise than “sooner”?
  • Figure 1 should be improved. The description of the Y-axis should be closer to the Y-axis. In the Legend, there is DI and NS that are not visible on the graph. The description of the X-axis is wrong. A, B and C are SPT-IDT+, and D and E are SPT+IDT+.
  • As a general comment, the title of a figure should be more descriptive and could indicate the main outcome. Then, following the title, write a brief description of what was done (“Comparison…”).
  • Figure 3’s title is bad and the font is different. The description of figure 3 lacks explanations (these are data from the 371 patients).
  • What are the 2 lines (323-324) connected to?
  • When a first author is mentioned (such as Jacobsen), “Jacobsen et al. (36)” should be written.
  • Line 374, there is “Pg. S6”. Where is it?
  • The Discussion is too long and include a new figure! Except for a summary figure, all figures should be in Results. When reading the Discussion, we are losing the focus of the study.

Author Response

Thanks you for your observations. Hopefully, following your concerns we have improved the methods and results sections to your satisfaction

Round 2

Reviewer 1 Report

the manuscript has been
sufficiently improved to warrant publication in Cells

Reviewer 2 Report

Thank you for responding to my concerns.